

# Effective sentence-level relation extraction model using entity-centric dependency tree

Seongsik Park[1] and Harksoo Kim[2]

[1] Department of Artificial Intelligence, Konkuk University, Seoul, Republic of South Korea
[2] Department of Computer Science and Engineering, Konkuk University, Seoul, Republic of South Korea

## ABSTRACT

The syntactic information of a dependency tree is an essential feature in relation extraction studies. Traditional dependency-based relation extraction methods can be categorized into hard pruning methods, which aim to remove unnecessary information, and soft pruning methods, which aim to utilize all lexical information. However, hard pruning has the potential to overlook important lexical information, while soft pruning can weaken the syntactic information between entities. As a result, recent studies in relation extraction have been shifting from dependency-based methods to pre-trained language model (LM) based methods. Nonetheless, LM-based methods increasingly demand larger language models and additional data. This trend leads to higher resource consumption, longer training times, and increased computational costs, yet often results in only marginal performance improvements. To address this problem, we propose a relation extraction model based on an entity-centric dependency tree: a dependency tree that is reconstructed by considering entities as root nodes. Using the entity-centric dependency tree, the proposed method can capture the syntactic information of an input sentence without losing lexical information. Additionally, we propose a novel model that utilizes entity-centric dependency trees in conjunction with language models, enabling efficient relation extraction without the need for additional data or larger models. In experiments with representative sentence-level relation extraction datasets such as TACRED, Re-TACRED, and SemEval 2010 Task 8, the proposed method achieves F1-scores of 74.9%, 91.2%, and 90.5%, respectively, which are state-of-the-art performances.

# INTRODUCTION

Relation extraction denotes the process of extracting triples (*i.e.,* the fundamental components of knowledge) from unstructured texts. The knowledge extracted from triples can be utilized for various natural language processing tasks such as knowledge base completion (*Shi & Weninger, 2018*; *Zhang et al., 2021*), knowledge discovery (*Bosselut et al., 2019*; *Zhang & Han, 2022*), question answering (*Abujabal et al., 2018*; *Zheng et al., 2019*), and dialogue systems (*Kim, Kwon & Kim, 2020*; *Young et al., 2018*). Sentence-level

Corresponding author
Harksoo Kim, nlp-drkim@konkuk.ac.kr

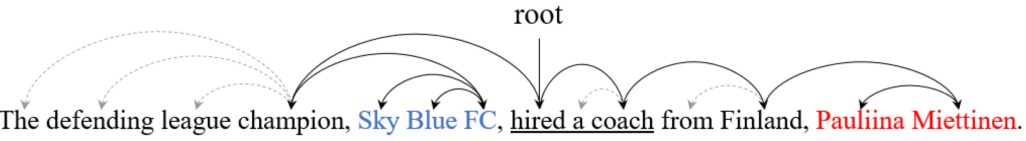

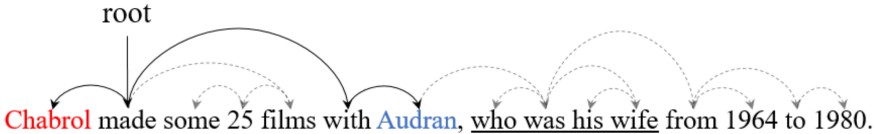

Figure 1 **An example of the sentence dependency tree.** The red and blue fonts represent the subject and object entities, respectively. The solid arcs represent the shortest dependency path between entities. The underlined phrases represent crucial information for relation extraction.

relation extraction involves classifying the relation between given entities (*i.e.,* subject and object entities) within a sentence (*Alt, Hübner & Hennig, 2019*; *Park & Kim, 2020*). Traditional relation extraction studies have employed various methods that utilize syntactic information between entities (*Bunescu & Mooney, 2005*; *Geng et al., 2020*; *Wang et al., 2023*). Specifically, methods that utilize subtrees, such as shortest dependency path (SDP) and the lowest common ancestor (LCA), help in extracting syntactic relation information between entities expressed in a sentence more effectively (*Zhang et al., 2022*; *Li et al., 2022*). However, this approach has the drawback of overlooking important lexicons or phrases that describe the relationship between the entities (*Guo, Zhang & Lu, 2019*).

Figure 1 shows examples of dependency trees for sentences that contain triples. For example, in the first sentence, {*Pauliina Miettinen*, employee_of, *Sky Blue FC*} is represented by the SDP of the two entities. In this case, we can easily extract the relation between entities using only SDP. However, in the second sentence, crucial information for relation extraction is lost when only SDP are utilized. The phrase "who was his wife" must be considered to extract the triple {*Chabrol*, spouse, *Audran*}. In this case, the use of SDP alone might impede successful relation extraction. To address this issue, recent dependency-based studies propose methods to maximize the utilization of all information in the dependency tree. The most representative method is the soft pruning strategy (*Guo, Zhang & Lu, 2019*; *Zhang et al., 2024*). Methods that utilize only the subtree between entities are categorized as hard pruning strategies, which exclude nodes not included in the subtree. In contrast, the soft pruning strategy employs an attention mechanism to assign weights to all nodes, allowing the model to adjust these weights during the learning phase to focus on information crucial for relation extraction. This approach minimizes the loss of information in sentences and facilitates effective relation extraction. However, it has the drawback of potentially weakening the important syntactic information between entities.

Since the introduction of pre-trained language models (LM), studies on relation extraction have primarily focused on leveraging the rich contextual representations provided by these language models (*Devlin et al., 2019*; *Alt, Hübner & Hennig, 2019*; *Xu et*

*al., 2022*). Language models enhance token representations with contextual information by using massive text corpora for self-supervised learning. Some studies propose relation extraction methods that focus on training language models with additional data specific to entities or using larger-scale language models (*Wan et al., 2023*; *Wadhwa, Amir & Wallace, 2023*; *Wang et al., 2022a*). While LM-based methods have shown superior performance over traditional dependency-based methods, this does not mean that the importance of syntactic information can be disregarded. The syntactic relationships between two entities can provide important information that language models alone may not capture. Ignoring this and merely increasing the size of the models or the training data can lead to additional overhead in terms of resource consumption and computational load.

This article proposes a method to resolve the trade-off between syntactic information and lexical information, which is a common issue in traditional dependency-based methods. Additionally, we demonstrate that by integrating this approach with pre-trained language models, syntactic information remains crucial for LM-based methods. First, we transform the dependency tree into an entity-centric dependency tree by designating an entity as the root node and reconstructing the tree based on the original dependency tree's structure. The entity-centric dependency tree is a structure in which the entity is connected to all words, and the syntactic distance between the entity and each word is represented as the label on the edges. Since sentence-level relation extraction involves two entities, an entity-centric dependency tree is generated for each of them. The root node of the dependency tree or the LCA node between entities does not necessarily explain the relationship between the entities. However, by changing the root node to the entity, it becomes easier to capture information that explains the relationship between the entities. The entity-centric dependency tree is used to calculate lexical information that is strongly related to the entities through an attention mechanism. The syntactic distance assigned to the edges is utilized to learn the syntactic relationships between the entities. These two processes are performed in parallel, enabling the extraction of relations using both lexical information and syntactic information, unlike traditional dependency-based methods. Furthermore, by integrating this approach with pre-trained language models, the performance of LM-based relation extraction is improved. The proposed method demonstrates comparable or superior relation extraction performance when compared to methods using additional data or larger models.

In experiments conducted using popular datasets for sentence-level relation extraction, such as TACRED, Re-TACRED, and SemEval-2010, we demonstrate the superiority of the proposed method. Our main contributions are as follows:

- We propose a method for transforming the original dependency tree into the entity-centric dependency tree that is better suited for relation extraction. By utilizing the entity-centric dependency tree, we can obtain both lexical and syntactic information that expresses the relationships between entities. Compared to previous studies using the original dependency tree, the entity-centric dependency tree demonstrates superior performance in relation extraction.

- The proposed method demonstrates performance that is similar to or better than relation extraction studies using more resources, even when using relatively fewer resources. Recent relation extraction studies achieve high performance by leveraging additional training data or larger language models. Our experiments show that the proposed method can achieve performance comparable to state-of-the-art models by using only dependency trees.

## RELATED WORK

Previous studies have proposed various methods to utilize the syntactic information provided by dependency trees in neural network models. One such method is hard pruning, which focuses on subtrees between entities, such as SDP and LCA, to remove unnecessary noise from sentences and utilize only essential information for relation extraction. *Xu et al. (2015a)* addressed the issue of irrelevant information in neural network models when subject and object entities are far apart by employing a convolutional neural network to learn robust relation representations from the SDP. *Miwa & Bansal (2016)* proposed an end-to-end model that encodes subtrees pruned around the LCA using a recurrent neural network to jointly extract entities and relations. *Zhang, Qi & Manning (2018)* improved the pruning method centered on the SDP and used it as input for a graph convolutional network to capture information necessary for relation extraction. However, these subtree-centric hard pruning methods have the drawback of potentially removing lexical or phrasal information that is crucial for relation extraction.

To mitigate these challenges, recent studies have proposed applying soft pruning to dependency-based methods. Soft pruning, which leverages attention mechanisms, enables the model to automatically learn the information necessary for relation extraction. *Guo, Zhang & Lu (2019)* was the first to propose a relation extraction method utilizing soft pruning. *Wang et al. (2023)* proposed a method to build a deep GCN structure by encoding adjacency matrices constructed through soft pruning into residual and densely connected paths. Similarly, *Zhang et al. (2024)* demonstrated performance improvements by calculating key information based on soft pruning and performing this in parallel with contextual encoding. However, the soft pruning method has the drawback of potentially weakening syntactic information, as it relies solely on the attention mechanism to calculate the key information between entities.

Since the advent of pre-trained language models (*Devlin et al., 2019*; *Liu et al., 2019*), most studies in relation extraction has focused on language models. Language models are capable of acquiring high-quality contextual representations because they are trained on large-scale corpora using various self-supervised learning strategies. *Alt, Hübner & Hennig (2019)* demonstrated that using only the contextual representations from pre-trained language models, without additional features, can achieve higher relation extraction performance than dependency-based methods. Similarly, *Soares et al. (2019)* proposed the most effective entity marking method for relation extraction by conducting comparative experiments that varied the input forms of entities in language models without using additional features. Beyond leveraging contextual representations from language models,

methods have also been proposed to extract key information provided by entities through additional data and pre-training. *Yamada et al. (2020)* conducted additional pre-training using a large entity-annotated corpus collected from Wikipedia to imbue the contextual representations of language models with the semantic representations of entities. *Wang et al. (2022a)* proposed a new language model with 10 billion parameters that enhances the structural understanding capabilities of language models by using task-agnostic corpora for structure generation pre-training. Recently, with the increasing prominence of large language models (LLMs) such as ChatGPT (*Brown et al., 2020*; *Touvron et al., 2023*) in various natural language processing fields, relation extraction methods leveraging these models have also been proposed. LLMs refer to language models that are trained on large-scale corpora, possess a massive number of parameters, benefit from human feedback, and are trained with instructions that equip them with human-like reasoning and problem-solving abilities. *Wan et al. (2023)* proposed a method to maximize the in-context learning capabilities of LLMs for relation extraction. They enhanced the relation extraction abilities of LLMs by selecting few-shot demonstrations from training data through a retrieval model and providing these as prompts. Similarly, *Wadhwa, Amir & Wallace (2023)* used few-shot prompts with LLMs and additionally applied the chain of thought (CoT) (*Wei et al., 2022*), achieving high performance. However, such LM-based methods have the drawback that utilizing more data and larger models linearly increases resource consumption and inference time. Additionally, the performance improvement is not proportional to the massive resources and computational effort invested.

We address the challenge of leveraging both syntactic and lexical information in dependency-based methods by designing an entity-centric dependency tree. The proposed method can selectively focus on words strongly related to the entity through the entity-centric dependency tree and can encode entity-centered syntactic distances to utilize the syntactic relationships between two entities. Additionally, when used with relatively smaller language models, our approach demonstrates comparable or superior results compared to models trained with larger datasets and more parameters. This suggests that researching the selection of appropriate features and their effective representation methods can enable more efficient relation extraction rather than merely pursuing larger models.

# METHODOLOGY

In this section, we describe the process of reconstructing entity-centric dependency trees and explain how these trees were applied to our relation extraction model.

## Task definition

In this article, we focus on sentence-level relation extraction. Let $T = \{w_0, w_1, ..., w_{|T|}\}$ denote the input words in a sentence and $e_{sbj}, e_{obj} \in T$ denote the subject and object entity spans. When given a predefined set of relation categories denoted as $R$, the objective of relation extraction is to predict the relation $r \in R$ between entities within the sentence.

## Entity-centric dependency tree

Syntactic information of a dependency tree is an essential feature for relation extraction. However, as observed in Fig. 1, hard pruning centered on the subtree between entities can

result in the loss of crucial lexical information. To address this problem, we reconstruct dependency trees in an entity-centric manner. Since there are two types of entities in a sentence (*i.e.,* subject and object entities), we create a separate dependency tree for each entity. Algorithm 1 describes the process of constructing an entity-centric dependency tree.

---

**Algorithm 1** Build Entity-centric Dependency Tree

**Input**: words $T = \{w_0, w_2, ..., w_t\}$
        entity span $e = \{w_1^e, w_2^e, ..., w_{|e|}^e\}$
        dependency tree $D$
**Output**: Entity-centric dependency tree $D_e$

1:  Initialize entity-centric dependency tree with $D_e \leftarrow \{root \xrightarrow{root} e\}$
2:  **for** $i \leftarrow 0$ **to** $|T|$ **do**
3:      **if** $w_i \notin e$ **then**
4:          Initialize $K_e$ with empty set
5:          **for** $w_j \in e$ **do**
6:              $K_e \leftarrow K_e \cup distance(w_i, w_j)$
7:          **end for**
8:          $k \leftarrow min(K_e)$
9:          $D_e \leftarrow D_e \cup \{e \xrightarrow{con:k} w_i\}$
10:     **end if**
11: **end for**
12: **return** $D_e$

---

In Algorithm 1, $T$ denotes a set of all words in a sentence, $e$ denotes the current entity span, and $D$ denotes a dependency tree of the sentence. The outputs $D_e$ denote the entity-centric tree reconstructed from the original dependency tree $D$. First, we set the entity as the root of the entity-centric dependency tree (line 1). Next, we traverse all words except for the entity and calculate a syntactic distance between the words and the entity (lines 2-11). Since the entity can consist of multiple words, we calculate the distance between the current word $w_i$ and all words comprising the entity (lines 5-7). $K_e$ represents temporary lists for storing the distance between words comprising the entity and $w_i$. The function $distance(w_i, w_j)$ denotes the distance in the original dependency tree $D$ between $w_i$ and $w_j$. For example, in the second sentence of Fig. 1, the entity 'Audran' can reach the word 'wife' through two hops in the dependency tree. After calculating the distance to all words comprising the entity, we consider the minimum value as the distance between the current word $w_i$ and the entity (line 8). This distance determines the labels of each word with respect to the entity in the entity-centric dependency tree (line 9). Figure 2 illustrates how a general dependency tree is reconstructed as an entity-centric dependency tree. In the new tree, dependency relations are replaced with syntactic distances. Next, the entity-centric dependency tree is inputted to the graph attention network, and entity-centric lexical information is extracted by soft pruning.

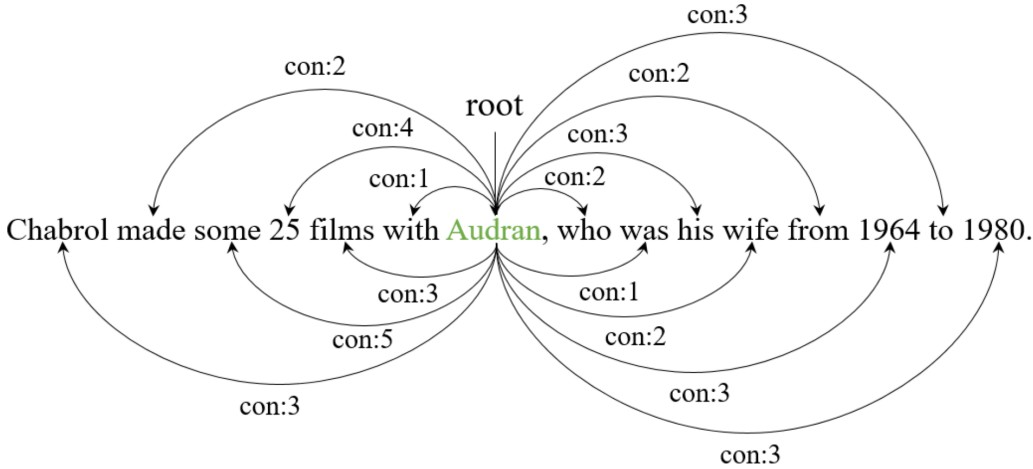

**Figure 2** **An example of the entity-centric dependency tree.** The green font denotes the entity, which is the center point of tree reconstruction (root node). The dependency relations are replaced with labels indicating the syntactic distance between the centric entity and each word.

## Relation extraction model

Figure 3 illustrates the overall architecture of the relation extraction model based on an entity-centric dependency tree. Most recent studies in natural language processing have used transformer-based pre-trained language models to extract contextual knowledge from given sentences or documents. We encoded each word in the sentences using transformer-based pre-trained language models. Once each token has been encoded by the language model, the average of the token encoding vectors corresponding to the entity is used as the entity encoding vector. For example, when $k$ and $l$ denote the start and end indices of the entity, the entity encoding vector $h^e$ is as follows:

$$h^e = \frac{1}{l-k+1}\sum_{i=k}^{l}h_i \tag{1}$$

where $h_i$ is $i$-th word encoding vector obtained from pre-trained language model.

Edges and syntactic distances in an entity-centric dependency tree are input into two separate networks. Edges provide entity-oriented lexical information in a graph attention network, while syntactic distances provide syntactic information between two entities. Entity-centric dependency trees connect every word node to the root node (entity). Therefore, in the graph attention network, attention between the entity and the word encoding vectors is used to perform soft pruning. The graph attention network is as follows:

$$g^e = \sum_{i\in\mathbb{N}_e}\alpha_i^e h_i \tag{2}$$

$$\alpha_i^e = \frac{\exp(s_i^e)}{\sum_{j\in\mathbb{N}_e}\exp(s_j^e)} \tag{3}$$

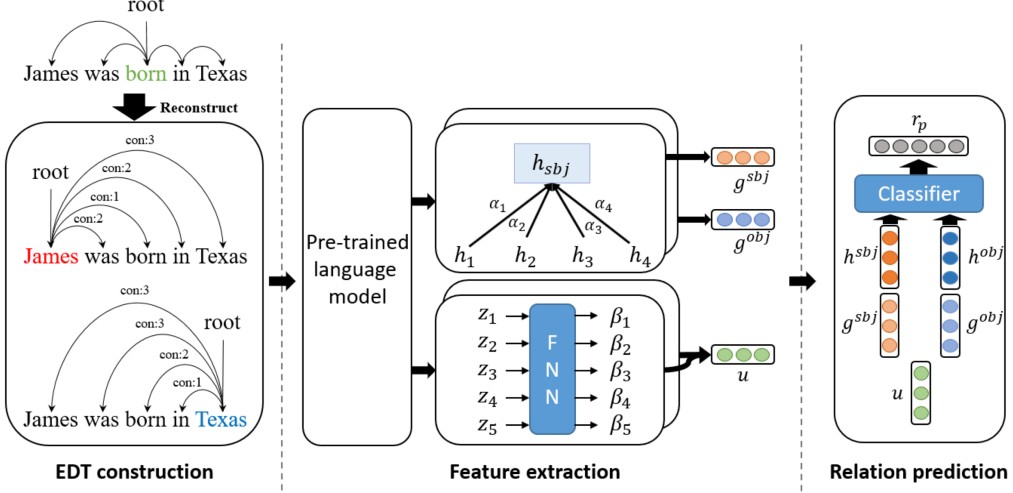

**Figure 3** **The overall architecture of the EDT based relation extraction model.**

$$s_i^e = \frac{h^e \cdot W_\alpha \cdot h_i}{\sqrt{d}} \tag{4}$$

where $g^e$ is the result of entering a dependency tree into a graph attention network, with entity $e$ as the root. Here, $\mathbb{N}e$ denotes the adjacent nodes (*i.e.,* all tokens except for the entity) of entity $e$, and $\alpha_i^e$ represents the attention weight between the entity $e$ and its $i$-th adjacent node. The attention score $s^e$ is calculated with bilinear attention, as expressed in Eq. (4), where $W_\alpha \in \mathbb{R}^{d \times d}$ is the trainable bilinear weight matrix. The attention score $s^e$ is normalized by the hidden size $d$. Lexical information for relation extraction is obtained in this process.

The syntactic distance is embedded with random initialization. These embeddings are passed through another attention layer, yielding syntactic information between two entities. The attention layer can be expressed as follows:

$$u = \sum_{t=0}^{T} \beta_t h_t \tag{5}$$

$$\beta_i = \frac{\beta_i^{sbj} \times \beta_i^{obj}}{\sum_{t=0}^{T} (\beta_t^{sbj} \times \beta_t^{obj})} \tag{6}$$

$$\beta_i^e = \frac{\exp(s_i^e)}{\sum_{t=0}^{T} \exp(s_t^e)} \tag{7}$$

$$s_i^e = \sigma(z_i^e W_\beta + b_\beta) \tag{8}$$

where $s_i^e$ is calculated using $z_i^e$ as input to a feed-forward neural network (FNN) layer. $z_i^e$ is the embedding vector for the syntactic distance between entity $e$ and the $i$th word. $W_\beta \in \mathbb{R}^{d_z \times 1}$ is a trainable parameter, and $d_z$ is the dimension of the syntactic distance embedding. $\sigma$ is an activation function such as ReLU (*Agarap, 2018*). Entity-centric dependency trees are created for both subject and object entities. Equation (6) illustrates combining the attention weights of both trees. $\beta_i^{sbj}$ and $\beta_i^{obj}$ denote the syntactic attention weights of the subject and object entities, respectively. The co-attention weight $\beta_i$ reflects common syntactic information between entities. $u$ is the distance attention output vector calculated from $\beta$. Through this process, we acquire syntactic information between the entities.

The final entity encoding vector and relation prediction are expressed as follows:

$$o^e = \sigma(W_e[h^e; g^e] + b_e) \tag{9}$$

$$r_p = \sigma(W_r[o^{sbj}; o^{obj}; u] + b_r) \tag{10}$$

where $o_e$ is the encoding vector of an entity and is calculated by FNNs with concatenations of entity embedding $h^e$ and graph attention network output $g^e$. The relation prediction $r_p$ is calculated by FNNs with concatenations of subject encoding $o^{sbj}$, object encoding $o^{obj}$, and common syntactic information $u$. Here, $W_e \in \mathbb{R}^{2d \times d}$ and $W_r \in \mathbb{R}^{3d \times |R|}$ are a trainable weight matrix. The $|R|$ represents the size of the set of relations. In relation extraction, predicting negative labels (*i.e.,* entity pairs without any relation) is equally crucial as predicting positive labels (*i.e.,* entity pairs with relation). To effectively differentiate them, we train the proposed method using the adapted thresholding loss (ATL) of *Zhou et al. (2021)*. The ATL is as follows:

$$L_p = -\sum_{r_p \in P_T} \frac{\exp(logit_{r_p})}{\sum_{r' \in P_T \cup \{TH\}} \exp(logit_{r'})} \tag{11}$$

$$L_n = -\log\left(\frac{\exp(logit_{TH})}{\sum_{r' \in N_T \cup \{TH\}} \exp(logit_{r'})}\right) \tag{12}$$

$$L = L_p + L_n \tag{13}$$

where $L_p$ is a loss function between positive label logit and a threshold logit, and $L_n$ is a loss function between negative label logit and a threshold logit. Through $L_p$ and $L_n$, the threshold value is trained to be lower than the logit values of positive labels and higher than the logit values of negative labels.

# EXPERIMENTS

## Dataset and evaluation measure

To evaluate the proposed method, we select TACRED (*Zhang et al., 2017*), Re-TACRED (*Stoica, Platanios & Póczos, 2021*) and SemEval 2010 Task 8 dataset (*Hendrickx et al., 2010*).

**TACRED**: TACRED is a popular large annotated dataset for sentence-level relation extraction, used in the Text Analysis Conference Knowledge Base Population challenge (TAC-KBP). TACRED covers 42 relation types (including the 'no_relation' class) and 16 entity types. The dataset is split into train, development, and evaluation sets, with 68,057, 22,630, and 15,509 sentences, respectively. In total, 78.4% of the data consists of negative samples. The TACRED dataset includes dependency trees analyzed with the Stanford parser. Our method creates entity-centric dependency trees from the given dependency tree.

**Re-TACRED**: While TACRED is a dataset constructed through human annotation, it includes approximately 25% incorrect labels. Re-TACRED is a significantly improved version of TACRED, involving the pruning of poorly annotated sentences and the resolution of ambiguities in TACRED relation definitions. Re-TACRED encompasses a total of 40 relation types. The dataset includes 58,465 sentences for training, 19,584 sentences for development, and 13,418 sentences for evaluation.

**SemEval 2010 Task 8 dataset**: Similar to TACRED, the SemEval 2010 Task 8 dataset is designed for sentence-level relation extraction. It is a publicly available relation extraction dataset with less data compared to TACRED. The training and test sets consist of 8,000 sentences and 2,717 sentences, respectively. There are a total of 19 relation types to classify, including the "other" class. Unlike TACRED, dependency trees are not provided. Therefore, we leverage the Stanford parser (https://nlp.stanford.edu/software/nndep.html) for dependency analysis and utilize the information to create entity-centric dependency trees.

The Stanford parser achieves an unlabeled attachment score (UAS) of 91.7% and a labeled attachment score (LAS) of 89.5% on the Penn Treebank dataset (*Marcus, Marcinkiewicz & Santorini, 1993*). Therefore, it is important to note that the experimental results may contain errors introduced by the Stanford parser. Table 1 describes statistics about TACRED, Re-TACRED and SemEval 2010 Task 8 dataset.

To evaluate the experimental results, we adopt the standard micro precision (P), recall (R), and F1-score (F1). All experimental results are obtained by averaging the results over five independent trials.

$$P = \frac{\# \ of \ correct \ predict}{\# \ of \ all \ triple \ in \ the \ model \ predict}$$

$$R = \frac{\# \ of \ correct \ predict}{\# \ of \ all \ triple \ in \ the \ dataset} \tag{14}$$

$$F1 = \frac{2PR}{P+R}.$$

## Implementation details

We conduct experiments with three popular pre-trained language models: BERT (*Devlin et al., 2019*), ELECTRA (*Clark et al., 2019*), and RoBERTa (*Liu et al., 2019*). The base-scaled

**Table 1   Statistics about TACRED, Re-TACRED and SemEval 2010 Task 8.**

|  | TACRED | Re-TACRED | SemEval |
|---|---|---|---|
| Length of sentence (min/max /avg) | 6/100/41.0 | 6/100/40.0 | 5/87/ 19.2 |
| # of relation types | 42 | 40 | 19 |
| # of entity types | 23 | 23 | - |
| # of samples (train/dev/test) | 68K/22K/15K | 58K/19K/13K | 8K/-/2.7K |
| # of positive/negative samples | 21K/84K (22%/78%) | 33K/57K (37%/63%) | 8.8K/2K (82%/18%) |

**Table 2   Performance comparison according to whether or not the entity-centric dependency tree is applied on TACRED.** "w/o EDT" refers to the performance with all layers related to EDT removed (GAT, SDA), while "w/ EDT" refers to after applying EDT.

| Models | F1 (%) | |
|---|---|---|
|  | w/o EDT | w/ EDT |
| BERT | 69.6 | 70.2 |
| ELECTRA | 71.6 | 72.0 |
| RoBERTa | 71.5 | 73.2 |

model consists of 12 transformer layers, with a hidden size of 768 and 110M trainable parameters. The large-scaled model consists of 24 transformer layers, with a hidden size of 1,024 and 340 M trainable parameters. The size of the syntactic distance embedding is set to be the same as the hidden size of the transformer. We apply a dropout of 0.1 to all the transformer and FNN layers. We select a learning rate of 3e-5 for the transformer layers and 1e-4 for the other layers. We use AdamW (*Loshchilov & Hutter, 2018*) to optimize the model.

## Experimental results

Table 2 illustrates the relation extraction performance improvement from applying entity-centric dependency trees (EDT) to three of the most popular pre-trained language models: BERT (*Devlin et al., 2019*), ELECTRA (*Clark et al., 2019*), and RoBERTa (*Liu et al., 2019*). All three models use the base scale model (110M). By using EDT, all models show performance improvements ranging from 0.4 to 1.7 points, with RoBERTa exhibiting the most noticeable performance improvement. Language models can enhance the contextual understanding of input sentences by pre-training on massive amounts of text data. Nevertheless, the performance observed with EDT indicates that EDT provides information beyond what language models can acquire alone. This demonstrates that the syntactic and lexical information from EDT has a significant effect on extracting relationships between entities. Since the performance improvement is most pronounced with RoBERTa, comparisons with other models are conducted using the combination of RoBERTa and EDT.

We compare the results of the proposed method to those of previous models using the TACRED, Re-TACRED, and SemEval 2010 Task 8 datasets. The models used for

comparison are the nine dependency-based methods and 15 LM-based methods listed below.

**Dependency-based methods**: C-GCN (Contextualized Graph Convolutional Network) (*Zhang, Qi & Manning, 2018*), AGGCN (Attention Guided Graph Convolutional Network) (*Guo, Zhang & Lu, 2019*), BT-LSTM (Bidirectional Tree-structured Long Short-term Memory) (*Geng et al., 2020*), DPRI (Data Partition and Representation Integration) (*Zhao et al., 2021*), GMAN (Gated Multi-window Attention Network) (*Xu et al., 2022*), DDT-REM (Relation Extraction Model with Dual Dependency Trees) (*Li et al., 2022*), MDR-GCN (Graph Convolutional Network with Multiple Dependency Representations) (*Hu et al., 2021*), ADPGCN (Attention Dual-Path Graph Convolutional Network) (*Wang et al., 2023*), DAGCN (Dual Attention Graph Convolutional Network) (*Zhang et al., 2024*)

**LM-based methods**: TRE (Transformer for Relation Extraction) (*Alt, Hübner & Hennig, 2019*), Indicator-aware (*Tao et al., 2019*), EPGNN (Entity Pair Graph based Neural Network) (*Zhao et al., 2019*), Entity-aware (*Wang et al., 2019*), ERNIE (Enhanced language RepresentatioN with Informative Entities) (*Zhang et al., 2019*), KnowBERT (Knowledge enhanced Bidirectional Encoder Representation from Transformers) (*Peters et al., 2019*), BERT-MTB (Bidirectional Encoder Representation from Transformers - Matching The Blanks) (*Soares et al., 2019*), CP (*Peng et al., 2020*), SpanBERT (Span Bidirectional Encoder Representation from Transformers) (*Joshi et al., 2020*), DeNERT-KG (Compound of Deep Q-Network, Named Entity Recognition, BERT, and Knowledge Graph) (*Yang, Yoo & Jeong, 2020*), LUKE (Language Understanding with Knowledge-based Embeddings) (*Yamada et al., 2020*), Typed-marker (*Zhou & Chen, 2022*), RECENT (RElation Classification with ENtity Type restriction) (*Lyu & Chen, 2021*), DeepStruct (*Wang et al., 2022a*), GPT-RE (Chat Generative Pre-trained Transformers (ChatGPT) based Relation Extraction) (*Wan et al., 2023*).

Table 3 shows the performance of the proposed method compared to other existing models on TACRED. In dependency-based methods, CGCN and AGGCN are representative models using hard pruning and soft pruning, respectively. When comparing the two models, AGGCN demonstrates slightly better performance, with a 0.8 point improvement. This suggests that preserving lexical information between entities through soft pruning is more effective for relation extraction than resolving long-distance dependencies between entities *via* hard pruning. The proposed method with EDT significantly outperforms AGGCN and exceeds the performance of all dependency-based methods. This indicates that the proposed method effectively utilizes both syntactic and lexical information for relation extraction by mitigating the trade-off between them. Additionally, the impact of the deep contextual representations provided by the language model is substantial. In fact, when comparing dependency-based methods with LM-based methods, most LM-based methods exhibit superior performance. Among LM-based methods, the proposed method stands out in terms of performance. Particularly, when compared with models using base-scale language models such as TRE, ERNIE, CP, Know-BERT, and DeNERT, the EDT RoBERTa-base model of the same scale achieves the highest performance. In some cases, our model outperforms the other models with extra training data which are ERNIE, LUKE, Know-BERT, DeNERT and BERT-MTB. Even

**Table 3 Performance comparison between EDT and previous models on TACRED.** A '+' mark indicates a model using extra training data. '†' and '††' marks indicate large and very large scaled models, respectively.

| Models | | P (%) | R (%) | F1 (%) |
|---|---|---|---|---|
| Dependency-based | ADPGCN | 72.9 | 63.6 | 67.9 |
| | MDR-GCN | 68.3 | 67.7 | 68.0 |
| | C-GCN | 71.3 | 65.4 | 68.2 |
| | DAGCN | 72.4 | 64.8 | 68.4 |
| | DDT-REM | 69.5 | 67.5 | 68.5 |
| | DPRI | 71.5 | 66.2 | 68.8 |
| | AGGCN | 73.1 | 64.2 | 69.0 |
| LM-based | TRE | 70.1 | 65.0 | 67.4 |
| | ERNIE[+] | 70.0 | 66.1 | 68.0 |
| | CP | – | - | 69.5 |
| | SpanBERT[†] | 70.8 | 70.9 | 70.8 |
| | KnowBERT[+] | 71.6 | 71.4 | 71.5 |
| | BERT-MTB[†+] | – | - | 71.5 |
| | GPT-RE[††] | – | - | 72.1 |
| | DeNERT[+] | 71.8 | 73.1 | 72.4 |
| | LUKE[†+] | 70.4 | 75.1 | 72.7 |
| | Typed-marker[†] | – | - | 74.6 |
| | RECENT[†] | **90.9** | 64.2 | 75.2 |
| | DeepStruct[††+] | – | – | **76.8** |
| EDT-based (ours) | RoBERTa-base | 72.7 | 73.6 | 73.2 |
| | RoBERTa-large[†] | 74.3 | **75.6** | 74.9 |

the EDT RoBERTa-base model outperforms LUKE by 0.5 points despite being smaller in scale. The LUKE model additionally fine-tunes the RoBERTa large model (340M) with a large amount of entity-annotated corpus obtained from Wikipedia. Furthermore, in terms of training costs, such as the amount of corpus and re-pretraining time, our model is more efficient since it only requires dependency parsing. However, We find that the proposed method shows lower performance than RECENT and DeepStruct. In the case of RECENT, several specific classifiers are trained based on combinations of entity types. Due to its strong dependency on entity types, it has the drawback of being challenging to use when data lacks entity types or when a new entity type appears in the data (*e.g.*, SemEval 2010 Task 8). In the case of DeepStruct, a model with 10 billion parameters is used, and additional data is also utilized. Therefore, the high performance of DeepStruct is due to the large parameter size and the use of extra data. In summary, while the proposed method requires a dependency tree, it is far more efficient compared to other models that utilize additional data and large-scale models. This indicates that, rather than merely using large datasets and models, it is essential to research effective ways to utilize appropriate features even in LM-based methods.

**Table 4  Performance comparison between EDT and previous models on Re-TACRED.**

| Models | | F1 (%) |
|---|---|---|
| Dependency-based | MDR-GCN | 79.8 |
| | C-GCN | 80.6 |
| | AGGCN | 80.8 |
| | DDT-REM | 82.4 |
| LM-based | SpanBERT[†] | 85.3 |
| | LUKE[†+] | 90.3 |
| | Typed-marker[†] | 91.1 |
| EDT-based (ours) | RoBERTa-large[†] | **91.2** |

Table 4 illustrates the performance comparison on Re-TACRED. The proposed method shows the best performance with an F1 score of 91.2%. When compared with dependency-based methods, it is evident that the proposed method achieves the highest performance, similar to the results on TACRED. Additionally, even when compared to LM-based methods using language models of the same scale, the proposed method still demonstrates superior performance. Compared with the experimental results on TACRED, the proposed method shows an F1 score improvement of 15.3 points. This is a larger improvement compared to the SpanBERT-large model's improvement of 14.5 points, indicating that the higher the integrity of the data, the better the efficiency of EDT.

Table 5 illustrates the performance comparison on the SemEval 2010 Task 8 dataset. Our model shows good performance on the SemEval 2010 Task 8 dataset as well. Among dependency-based methods, GMAN, like the proposed method, utilizes information from the dependency tree in conjunction with the language model. GMAN also demonstrates higher performance than most LM-based methods on the SemEval 2010 Task 8 dataset. Similar to experiments on other datasets, the proposed method shows higher performance than models that use additional datasets, such as KnowBERT and BERT-MTB. This further proves that increasing the size of the language model and the training data is not the only solution. Indicator-aware defines a syntactic indicator that is syntactically important in a sentence, which shows good performance in relation extraction. The syntactic indicator is manually designed based on the results of morphological analysis and named entity recognition. The attention distribution $\beta$ in Eq. (6) plays a similar role to this syntactic indicator. Unlike Indicator-aware, our proposed method automatically calculates $\beta$ through training. This is a further advantage of the proposed approach. Compared to TACRED, the performance of GPT-RE has improved significantly on SemEval. GPT-RE is a model based on GPT-3 (*Brown et al., 2020*). Considering the massive number of parameters in GPT-3 (175 billion parameters), it is clear that this approach lacks consistency and efficiency. The proposed method adopts the size of RoBERTa Large (340 million parameters). In terms of efficiency, it is evident that the proposed approach, even at roughly 1/500th the size, can achieve comparable performance, making it better suited for relation extraction.

**Table 5    Performance comparison between EDT and previous models on SemEval 2010 Task 8.**

| Models | | F1 (%) |
|---|---|---|
| Dependency-based | C-GCN | 84.8 |
| | DDT-REM | 84.9 |
| | MDR-GCN | 84.9 |
| | AGGCN | 85.7 |
| | ADPGCN | 85.9 |
| | DAGCN | 86.0 |
| | BT-LSTM | 87.1 |
| | GMAN | 90.3 |
| LM-based | TRE | 87.1 |
| | CP | 87.6 |
| | Entity-aware | 89.0 |
| | KnowBERT[+] | 89.1 |
| | BERT-MTB[†+] | 89.5 |
| | EPGNN | 90.2 |
| | Indicator-aware | 90.4 |
| | GPT-RE[††] | **91.9** |
| EDT-based (ours) | RoBERTa-large[†] | 90.5 |

**Table 6    An ablation study for the proposed method using TACRED.** GAT denotes the graph attention network for lexical information, SDA denotes the syntactic distance attention for syntactic information.

| Model | F1 (%) |
|---|---|
| EDT-RoBERTa-based model | 73.2 |
| - GAT | 72.9 |
| - SDA | 72.4 |
| - GAT, SDA | 71.5 |

## Analysis

To measure the effectiveness of each component in the proposed method, we conducted an ablation test, as presented in Table 6. We found that the GAT contributes to an improvement of 0.3 points, as it effectively acquires overall lexical information. The F1 score dropped by 0.8 points when we removed the SDA, resulting in an F1 score of 72.4%. This indicates that the SDA significantly contributes to acquiring syntactic information from the entity-centric dependency trees. When both GAT and SDA are removed, the performance drops by 1.7 points. This decline is greater than the sum of the declines caused by each component individually, suggesting that the two components not only complement each other but also have a synergistic effect. Although the pre-trained language models have abundant contextual knowledge, we observed that their performance can be further improved with additional lexical and syntactic information.

Table 7 presents the experimental results applying the representative hard pruning method, C-GCN, and the soft pruning method, AGGCN, to the RoBERTa-base model for a fair comparison between dependency-based methods and the proposed method. The 'w/o

**Table 7  Performance comparison based on syntactic information utilization methods on the TA-CRED dataset.** All models used the RoBERTa-base model as the encoder.

| Model | F1 (%) |
|---|---|
| w/o dependency tree | 71.5 |
| C-GCN | 71.9 |
| AGGCN | 72.2 |
| **EDT** | **73.2** |

dependency tree' indicates the performance of using only the RoBERTa-base model for relation extraction without utilizing a dependency tree. Compared to this baseline, C-GCN and AGGCN show performance improvements of 0.4 points and 0.7 points, respectively. The proposed method achieves the highest performance improvement with an increase of 1.7 points. The performance difference between C-GCN and AGGCN is attributed to the utilization of lexical information. Since C-GCN is based on hard pruning, it can overlook important lexical information crucial for relation extraction. In contrast, AGGCN, based on soft pruning, reduces the likelihood of this issue occurring. The proposed method has the advantage of harmoniously utilizing both lexical and syntactic information compared to the two methods. The performance gap shown in Table 7 is attributed to these characteristics.

To check whether syntactic information and lexical information are effectively learned through the proposed neural network architecture, we visualized the attention weights (*i.e.,* $\alpha + \beta$) of each token through heat maps, as shown in Fig. 4. In the case of the first sentence, syntactic information from words between 'Pauliina Miettien' and 'Sky Blue FC' is sufficient for relation extraction, and the attention weights are well focused on words that constitute the SDP of the two entities. However, the sole use of SDP does not provide enough information for relation extraction in the second sentence. Fortunately, the attention weights are distributed not only among the SDP but also among key phrases outside the SDP. Hence, the model was able to predict the spouse relationship accurately. Since TACRED is a semi-automatically annotated dataset, we observed instances with mislabeled target relation tags. In cases such as the third sentence, the proposed method predicted labels that are more appropriate for the ground-truth labels. The fourth sentence is an example of a typical off-target prediction by the proposed method. The attention weights are mainly distributed on words such as 'looked' (the LCA of the two entities), 'father,' and 'daughter' (information necessary for relation extraction). However, for relation extraction, the model should be able to locally reason that the subject entity 'his' refers to 'Knox's father' or 'Curt Knox,' and the object entity 'she' refers to 'his daughter.' Consequently, further study on complex information (*e.g.*, coreference resolution) and reasoning beyond lexical information and syntactic information is needed for complete relation extraction.

## CONCLUSION

In this article, we propose a relation extraction model using entity-centric dependency, which effectively incorporates the syntactic information of dependency trees and lexical information. The proposed method outperforms existing dependency-based methods

| Sentence | Predict | Correct |
|---|---|---|
| The defending league champion, Sky Blue FC, hired a coach from Finland, Pauliina Miettien. | Employee_of | Employee_of |
| Chabrol made some 25 films with Audarn, who was his wife from 1964 to 1980. | Spouse | Spouse |
| That too may have resonated with militants in that region, said Ahmed Rashid, a Lahore-based analyst and author of a book on the Taliban. | Title | no_relation |
| Knox's father, Curt Knox, said his daughter looked "confident in what she wants to say." | no_relation | Children |

**Figure 4** **Heat map visualizing attention weights.** The red and blue fonts denote the subject and object entities, respectively. The shade of the words represents the attention weight of $\alpha + \beta$. The darker the shade, the higher the attention weight.

on popular relation extraction datasets such as TACRED, Re-TACRED, and SemEval 2010 Task 8. This success can be attributed to the effective mitigation of the trade-off between lexical and syntactic information, a common challenge in traditional dependency-based methods. Furthermore, compared to LM-based methods, the proposed method demonstrates superior performance despite utilizing relatively smaller language models, outperforming methods that employ larger language models or additional training data. Simply increasing the size of parameters and data does not constitute an efficient approach when considering the resources consumed and the marginal performance gains achieved.

Our experiments demonstrate that leveraging entity-centric dependency trees enables efficient and robust sentence-level relation extraction. Future study will focus on extending the proposed method to document-level or multi-document relation extraction. Given that documents consist of multiple sentences, directly applying sentence-level graph structures like dependency trees is challenging. Therefore, we plan to explore entity-centric graphs applicable at the document level and investigate their use in relation extraction.

### Funding
This work was supported by Institute of Information & Communications Technology Planning & Evaluation (IITP) grant funded by the Korea government (MSIT) (RS-2023-00216011, Development of artificial complex intelligence for conceptually understanding and inferring like human). Also, this work was supported by Institute for Information & communications Technology Planning & Evaluation(IITP) grant funded by the Korea government(MSIT) (No. RS-2022-II220369, (Part 4) Development of AI Technology to support Expert Decision-making that can Explain the Reasons/Grounds for Judgment Results based on Expert Knowledge). The funders had no role in study design, data collection and analysis, decision to publish, or preparation of the manuscript.

### Grant Disclosures
The following grant information was disclosed by the authors:

Institute of Information & Communications Technology Planning & Evaluation (IITP) grant funded by the Korea government (MSIT): RS-2023-00216011.

Institute for Information & communications Technology Planning & Evaluation (IITP) grant funded by the Korea government (MSIT):  No. RS-2022-II220369.

## Competing Interests

The authors declare there are no competing interests.

## Author Contributions

- Seongsik Park conceived and designed the experiments, performed the experiments, analyzed the data, performed the computation work, prepared figures and/or tables, authored or reviewed drafts of the article, and approved the final draft.
- Harksoo Kim conceived and designed the experiments, authored or reviewed drafts of the article, and approved the final draft.

## Data Availability

The original data is available at Kaggle: https://www.kaggle.com/datasets/drtoshi/semeval2010-task-8-dataset?resource=download.

TACRED is available at Zhong, Victor, et al. TAC Relation Extraction Dataset LDC2018T24. Web Download. Philadelphia: Linguistic Data Consortium, 2018. https://doi.org/10.35111/m0kp-4w25.

The source code is available in the Supplemental File.

## Supplemental Information

Supplemental information for this article can be found online at http://dx.doi.org/10.7717/peerj-cs.2311#supplemental-information.

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
