# Peer review of "Effective sentence-level relation extraction model using entity-centric dependency tree"

_PeerJ Computer Science, doi:10.7717/peerj-cs.2311_

## Round 0.1 · original submission · Major Revisions

Dear authors,

The reviewers have commented on your paper. They indicated that it is not acceptable for publication in its present form.

However, if you feel that you can suitably address the reviewers' comments, I invite you to revise and resubmit your manuscript.

Please carefully address the issues raised in the comments.

If you are submitting a revised manuscript, please also:

a) outline each change made (point by point) as raised in the reviewer comments

b) provide a suitable rebuttal to each reviewer comment not addressed

**Language Note:** The review process has identified that the English language must be improved. PeerJ can provide language editing services - please contact us at [email protected] for pricing (be sure to provide your manuscript number and title). Alternatively, you should make your own arrangements to improve the language quality and provide details in your response letter. – PeerJ Staff

Reviewer 1 ·

Basic reporting

The authors show promise in their work, yet certain issues need addressing. The manuscript requires meticulous polishing of its English and writing, including extensive editing of vocabulary, language, syntax, phrasing, and punctuation. Simplify the abstract by using only the simple present tense.

Experimental design

-

Validity of the findings

The authors show promise in their work, yet certain issues need addressing. The manuscript requires meticulous polishing of its English and writing, including extensive editing of vocabulary, language, syntax, phrasing, and punctuation. Simplify the abstract by using only the simple present tense.
In the introduction, establish a clear connection between the current state of research and your paper's objectives. Provide a concise analysis of the state of the art, highlighting identified knowledge gaps and their relevance to your study. Justify the novelty and significance of your paper's goals, citing relevant previous studies. Clearly articulate the research gaps and contributions. Note that without a clear research gap and novelty, the paper may not progress further.
The introduction and literature survey are lacking. Incorporate recent articles to strengthen the literature review. Many pertinent research articles are overlooked in the current version; therefore, broaden the literature search to enhance the manuscript.
The methodological choices lack sufficient justification, and their impact on performance is not adequately analyzed. The manuscript lacks clarity regarding the primary innovation and value of the methodology. Provide a more comprehensive discussion of the pros and cons of the proposed methodology.
The experimental section requires improvement, including comparison with recent state-of-the-art methods. Enhance the conclusion by highlighting key achievements, managerial insights, and future directions.

Reviewer 2 ·

Basic reporting

This manuscript claims to present a new approach in relationship extraction studies. There are presumptions that he contributed to the literature in this field. In summary, it innovatively proposes to methodologically transform the traditional tree dependency structure into an “entity-centered dependency tree” that is more suitable for relationship extraction. This proposed methodology appears to be more effective in capturing both relationships between words and dependency information.

The article is well structured. The following comments can be accepted after being made.
1- The number of references should be increased. Comparison of the results obtained with the findings of similar studies in the past should be made multidimensionally and in more depth.
2-The original value and usability of the solution proposed in the study should be discussed in more detail.

Experimental design

The methodology used here basically creates entity-centered dependency trees. And it focuses on applying these trees to the relationship extraction model. This methodology aims to combine syntactic and lexical information, which is an important component of relationship extraction.

Some interesting methodological steps are suggested here;

First, the syntactic distance of each word to the entity is calculated (and this information is used to construct the entity-centered dependency tree). The generated entity-centered dependency trees are then fed into the relationship extraction model (This model obtained the encodings of each word through a series of operations.). Then, the tree structure and dependency distances were processed separately to be used in relationship extraction. Finally, the relationship extraction model makes relationship predictions using entity encodings (as well as syntactic information).

Yes, the methodology seems generally reasonable and this innovative voice sounds good. Because it basically brings together the important elements of the relationship extraction process. However, the effectiveness and general performance of the techniques used should be confirmed by repeating subsequent studies and the reader should be satisfied. This means that additional verifications may be required that all processes mentioned in the text are feasible and efficient.

Validity of the findings

This study seems to offer the reader a very comprehensive analysis. First, the datasets and evaluation metrics used are clearly defined. Thus, it can be said that the reliability of the study has increased. The methodology and application steps are explained. It also explains how each component works.

It is an exemplary and correct approach to present the results of the study by comparing it with previous models in order to show that the proposed method has a competitive performance and especially gives better results than others. The contribution of each component was also evaluated.

It seems that this study will make an interesting and important contribution. However, some parts of the work contain intense mathematical formulas. Also, I think that the accuracy of the data set labels and the generalization ability of the method should be further tested in future studies.

---

## Round 0.2 · accepted · Accept

Dear author,

After review of the revised version of the paper, both reviewers have accepted it. Congratulations.

Reviewer 1 ·

Basic reporting

-

Experimental design

-

Validity of the findings

-

Reviewer 2 ·

Basic reporting

This paper has been improved in terms of language, literature, references, figures, and tables. Satisfactory explanations have been added for the verification of the findings of the study.

Experimental design

This article is original research within the scope and purpose of the journal. The model is explained in a very informative way.

Validity of the findings

The potential impact and novelty of the study appear satisfactory. The results are logically stated. The purpose, context, and conclusions are well-founded.

Additional comments

no comment